# NeurWIN: Neural Whittle Index Network For Restless Bandits Via Deep RL

**Khaled Nakhleh**[1], **Santosh Ganji**[1], **Ping-Chun Hsieh**[2], **I-Hong Hou**[1], **Srinivas Shakkottai**[1]

[1] Electrical and Computer Engineering Department
Texas A&M University
College Station, TX
`{khaled.jamal, sant1, ihou, sshakkot}@tamu.edu`

[2] Department of Computer Science
National Chiao Tung University, Taiwan
`pinghsieh@nctu.edu.tw`

## Abstract

Whittle index policy is a powerful tool to obtain asymptotically optimal solutions for the notoriously intractable problem of restless bandits. However, finding the Whittle indices remains a difficult problem for many practical restless bandits with convoluted transition kernels. This paper proposes NeurWIN, a neural Whittle index network that seeks to learn the Whittle indices for any restless bandits by leveraging mathematical properties of the Whittle indices. We show that a neural network that produces the Whittle index is also one that produces the optimal control for a set of Markov decision problems. This property motivates using deep reinforcement learning for the training of NeurWIN. We demonstrate the utility of NeurWIN by evaluating its performance for three recently studied restless bandit problems. Our experiment results show that the performance of NeurWIN is significantly better than other RL algorithms.

## 1 Introduction

Many sequential decision problems can be modeled as multi-armed bandit problems. A bandit problem models each potential decision as an arm. In each round, we play $M$ arms out of a total of $N$ arms by choosing the corresponding decisions. We then receive a reward from the played arms. The goal is to maximize the expected long-term total discounted reward. Consider, for example, displaying advertisements on an online platform with the goal to maximize the long-term discounted click-through rates. This can be modeled as a bandit problem where each arm is a piece of advertisement and we choose which advertisements to be displayed every time a particular user visits the platform. It should be noted that the reward, i.e., click-through rate, of an arm is not stationary, but depends on our actions in the past. For example, a user that just clicked on a particular advertisement may be much less likely to click on the same advertisement in the near future. Such a problem is a classic case of the *restless bandit problem*, where the reward distribution of an arm depends on its state, which changes over time based on our past actions.

The restless bandit problem is notoriously intractable [20]. Most recent efforts, such as *recovering bandits* [21], *rotting bandits* [23], and *Brownian bandits* [24], only study some special instances of the restless bandit problem. The fundamental challenge of the restless bandit problem lies in the explosion of state space, as the state of the entire system is the Cartesian product of the states of individual arms. A powerful tool traditionally used to solve the RMABs' decision-making problem

35th Conference on Neural Information Processing Systems (NeurIPS 2021).

is the Whittle index policy [30]. In a nutshell, the Whittle index policy calculates a Whittle index for each arm based on the arm's current state, where the index corresponds to the amount of cost that we are willing to pay to play the arm, and then plays the arm with the highest index. When the indexability condition is satisfied, it has been shown that the Whittle index policy is asymptotically optimal in a wide range of settings.

In this paper, we present Neural Whittle Index Network (NeurWIN), a principled machine learning approach that finds the Whittle indices for virtually all restless bandit problems. We note that the Whittle index is an artificial construct that cannot be directly measured. Finding the Whittle index is typically intractable. As a result, the Whittle indices of many practical problems remain unknown except for a few special cases.

We are able to circumvent the challenges of finding the Whittle indices by leveraging an important mathematical property of the Whittle index: Consider an alternative problem where there is only one arm and we decide whether to play the arm in each time instance. In this problem, we need to pay a constant cost of $\lambda$ every time we play the arm. The goal is to maximize the long-term discounted net reward, defined as the difference between the rewards we obtain from the arm and the costs we pay to play it. Then, the optimal policy is to play the arm whenever the Whittle index becomes larger than $\lambda$. Based on this property, a neural network that produces the Whittle index can be viewed as one that finds the optimal policy for the alternative problem for any $\lambda$.

Using this observation, we propose a deep reinforcement learning method to train NeurWIN. To demonstrate the power of NeurWIN, we employ NeurWIN for three recently studied restless bandit problems, namely, recovering bandit [21], wireless scheduling [1], and stochastic deadline scheduling [34]. We compare NeurWIN against five other reinforcement learning algorithms and the application-specific baseline policies in the respective restless bandit problems. Experiment results show that the index policy using our NeurWIN significantly outperforms other reinforcement learning algorithms. Moreover, for problems where the Whittle indices are known, NeurWIN has virtually the same performance as the corresponding Whittle index policy, showing that NeurWIN indeed learns a precise approximation to the Whittle indices.

The rest of the paper is organized as follows: Section 2 reviews related literature. Section 3 provides formal definitions of the Whittle index and our problem statement. Section 4 introduces our training algorithm for NeurWIN. Section 5 demonstrates the utility of NeurWIN by evaluating its performance under three recently studied restless bandit problems. Finally, Section 6 concludes the paper.

## 2  Related work

Restless bandit problems were first introduced in [30]. They are known to be intractable, and are in general PSPACE hard [20]. As a result, many studies focus on finding the Whittle index policy for restless bandit problems, such as in [16, 8, 18, 26]. However, these studies are only able to find the Whittle indices under various specific assumptions about the bandit problems.

There have been many studies on applying RL methods for bandit problems. [9] proposed a tool called Uniform-PAC for contextual bandits. [35] described a framework-agnostic approach towards guaranteeing RL algorithms' performance. [13] introduced contextual decision processes (CDPs) that encompass contextual bandits for RL exploration with function approximation. [22] compared deep neural networks with Bayesian linear regression against other posterior sampling methods. However, none of these studies are applicable to restless bandits, where the state of an arm can change over time.

Deep RL algorithms have been utilized in problems that resemble restless bandit problems, including HVAC control [29], cyber-physical systems [17], and dynamic multi-channel access [28]. In all these cases, a major limitation for deep RL is scalability. As the state spaces grows exponentially with the number of arms, these studies can only be applied to small-scale systems, and their evaluations are limited to cases when there are at most 5 zones, 6 sensors, and 8 channels, respectively.

An emerging research direction is applying machine learning algorithms to learn Whittle indices. [6] proposed employing the LSPE(0) algorithm [32] coupled with a polynomial function approximator. The approach was applied in [3] for scheduling web crawlers. However, this work can only be applied to restless bandits whose states can be represented by a single number, and it only uses a polynomial function approximator, which may have low representational power [25].

[5] studied a public health setting and models it as a restless bandit problem. A Q-learning based Whittle index approach was formulated to ideally pick patients for interventions based on their states. [11] proposed a Q-learning based heuristic to find Whittle indices. However, as shown in its experiment results, the heuristic may not produce Whittle indices even when the training converges. [4] proposed WIBQL: a Q-learning method for learning the Whittle indices by applying a modified tabular relative value iteration (RVI) algorithm from [2]. The experiments presented here show that WIBQL does not scale well with large state spaces.

# 3 Problem Setting

In this section, we provide a brief overview of restless bandit problems and the Whittle index. We then formally define the problem statement.

## 3.1 Restless Bandit Problems

A restless bandit problem consists of $N$ restless arms. In each round $t$, a control policy $\pi$ observes the state of each arm $i$, denoted by $s_i[t]$, and selects $M$ arms to activate. We call the selected arms as *active* and the others as *passive*. We use $a_i[t]$ to denote the policy's decision on each arm $i$, where $a_i[t] = 1$ if the arm is active and $a_i[t] = 0$ if it is passive at round $t$. Each arm $i$ generates a stochastic reward $r_i[t]$ with distribution $R_{i,act}(s_i[t])$ if it is active, and with distribution $R_{i,pass}(s_i[t])$ if it is passive. The state of each arm $i$ in the next round evolves by the transition kernel of either $P_{i,act}(s_i[t])$ or $P_{i,pass}(s_i[t])$, depending on whether the arm is active. The goal of the control policy is to maximize the expected total discounted reward, which can be expressed as $\mathbb{E}_\pi[\sum_{t=0}^\infty \sum_{i=1}^N \beta^t r_i[t]]$ with $\beta$ being the discount factor.

A control policy is effectively a function that takes the vector $(s_1[t], s_2[t], \ldots, s_N[t])$ as the input and produces the vector $(a_1[t], a_2[t], \ldots, a_N[t])$ as the output. It should be noted that the space of input is exponential in $N$. If each arm can be in one of $K$ possible states, then the number of possible inputs is $K^N$. This feature, which is usually referred to as the curse of dimensionality, makes finding the optimal control policy intractable.

## 3.2 The Whittle Index

An index policy seeks to address the curse of dimensionality through decomposition. In each round, it calculates an index, denoted by $W_i(s_i[t])$, for each arm $i$ based on its current state. The index policy then selects the $M$ arms with the highest indices to activate. It should be noted that the index of an arm $i$ is independent from the states of any other arms. In this sense, learning the Whittle index of a restless arm is an *auxiliary task* to finding the control policy for restless bandits.

Obviously, the performance of an index policy depends on the design of the index function $W_i(\cdot)$. A popular index with solid theoretical foundation is the Whittle index, which is defined below. Since we only consider one arm at a time, we drop the subscript $i$ for the rest of the paper.

Consider a system with only one arm, and an activation policy that determines whether to activate the arm in each round $t$. Suppose that the activation policy needs to pay an activation cost of $\lambda$ every time it chooses to activate the arm. The goal of the activation policy is to maximize the total expected discounted net reward, $\mathbb{E}[\sum_{t=0}^\infty \beta^t(r[t] - \lambda a[t])]$. The optimal activation policy can be expressed by the set of states in which it would activate this arm for a particular $\lambda$, and we denote this set by $\mathcal{S}(\lambda)$. Intuitively, the higher the cost, the less likely the optimal activation policy would activate the arm in a given state, and hence the set $\mathcal{S}(\lambda)$ would decrease monotonically. When an arm satisfies this intuition, we say that the arm is *indexable*.

**Definition 1** (Indexability). *An arm is said to be indexable if $\mathcal{S}(\lambda)$ decreases monotonically from the set of all states to the empty set as $\lambda$ increases from $-\infty$ to $\infty$. A restless bandit problem is said to be indexable if all arms are indexable.*

**Definition 2** (The Whittle Index). *If an arm is indexable, then its Whittle index of each state $s$ is defined as $W(s) := \sup_\lambda\{\lambda : s \in \mathcal{S}(\lambda)\}$.*

Even when an arm is indexable, finding its Whittle index can still be intractable, especially when the transition kernel of the arm is convoluted.[1] Our NeurWIN finds the Whittle index by leveraging the following property of the Whittle index: Consider the single-armed bandit problem. Suppose the initial state of an indexable arm is $s$ at round one. Consider two possibilities: The first is that the activation policy activates the arm at round one, and then uses the optimal policy starting from round two; and the second is that the activation policy does not activate the arm at round one, and then uses the optimal policy starting from round two. Let $Q_{\lambda,act}(s)$ and $Q_{\lambda,pass}(s)$ be the expected discounted net reward for these two possibilities, respectively, and let $D_s(\lambda) := \big(Q_{\lambda,act}(s) - Q_{\lambda,pass}(s)\big)$ be their difference. Clearly, the optimal activation policy should activate an arm under state $s$ and activation cost $\lambda$ if $D_s(\lambda) \geq 0$. We present the property more formally in the following proposition:

**Proposition 1.** *[36, Thm 3.14] If an arm is indexable, then, for every state s, $D_s(\lambda) \geq 0$ if and only if $\lambda \leq W(s)$.*

Our NeurWIN uses Prop. 1 to train neural networks that predict the Whittle index for any indexable arms. From Def. 1, a sufficient condition for indexability is when $D_s(\lambda)$ is a decreasing function. Thus, we define the concept of *strong indexability* as follows:

**Definition 3** (Strong Indexability). *An arm is said to be strongly indexable if $D_s(\lambda)$ is strictly decreasing in $\lambda$ for every state s.*

Intuitively, as the activation cost increases, it becomes less attractive to activate the arm in any given state. Hence, one would expect $D_s(\lambda)$ to be strictly decreasing in $\lambda$ for a particular state $s$. In Section 5.5, we further use numerical results to show that all three applications we evaluate in this paper are strongly indexable.

## 3.3 Problem Statement

We now formally describe the objective of this paper. We assume that we are given a simulator of one single restless arm as a black box. The simulator provides two functionalities: First, it allows us to set the initial state of the arm to any arbitrary state $s$. Second, in each round $t$, the simulator takes $a[t]$, the indicator function that the arm is activated, as the input and produces the next state $s[t+1]$ and the reward $r[t]$ as the outputs.

Our goal is to derive low-complexity index algorithms for restless bandit problems by training a neural network that approximates the Whittle index of each restless arm using its simulator. A neural network takes the state $s$ as the input and produces a real number $f_\theta(s)$ as the output, where $\theta$ is the vector containing all weights and biases of the neural network. Recall that $W(s)$ is the Whittle index of the arm. We aim to find appropriate $\theta$ that makes $|f_\theta(s) - W(s)|$ small for all $s$. Such a neural network is said to be *Whittle-accurate*.

**Definition 4** (Whittle-accurate). *A neural network with parameters $\theta$ is said to be $\gamma$-Whittle-accurate if $|f_\theta(s) - W(s)| \leq \gamma$, for all s.*

# 4 NeurWIN Algorithm: Neural Whittle Index Network

In this section, we present NeurWIN, a deep-RL algorithm that trains neural networks to predict the Whittle indices. Since the Whittle index of an arm is independent from other arms, NeurWIN trains one neural network for each arm independently. In this section, we discuss how NeurWIN learns the Whittle index for one single arm.

## 4.1 Conditions for Whittle-Accurate

Before presenting NeurWIN, we discuss the conditions for a neural network to be $\gamma$-Whittle-accurate.

Suppose we are given a simulator of an arm and a neural network with parameters $\theta$. We can then construct an environment of the arm along with an activation cost $\lambda$ as shown in Fig. 1. In each round $t$, the environment takes the real number $f_\theta(s[t])$ as the input.

---

[1][19] described a generic approach for finding the Whittle index. The complexity of this approach is at least exponential to the number of states.

The input is first fed into a step function to produce $a[t] = 1(f_\theta(s[t]) \geq \lambda)$, where $1(\cdot)$ is the indicator function. Then, $a(t)$ is fed into the simulator of the arm to produce $r[t]$ and $s[t+1]$. Finally, the environment outputs the net reward $r[t] - \lambda a[t]$ and the next state $s[t+1]$. We call this environment $Env(\lambda)$. Thus, the neural network can be viewed as a controller for $Env(\lambda)$. The following corollary is a direct result from Prop. 1.

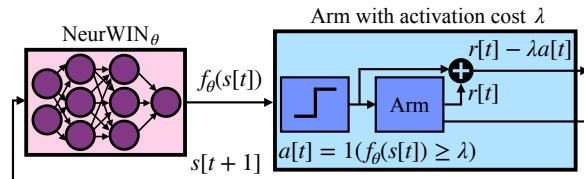

Figure 1: An illustrative motivation of NeurWIN.

**Corollary 1.** *If the arm is indexable and $f_\theta(s) = W(s), \forall s$, then the neural network with parameters $\theta$ is the optimal controller for $Env(\lambda)$, for any $\lambda$ and initial state $s[0]$. Moreover, given $\lambda$ and $s[0]$, the optimal discounted net reward is $\max\{Q_{\lambda,act}(s[0]), Q_{\lambda,pass}(s[0])\}$.*

Corollary 1 can be viewed as a necessary condition for a neural network to be 0-Whittle-accurate. Next, we establish a sufficient condition that shows how a *near-optimal* neural network controller for $Env(\lambda)$ is also Whittle-accurate. Let $\tilde{Q}_\theta(s, \lambda)$ be the average reward of applying a neural network with parameters $\theta$ to $Env(\lambda)$ with initial state $s$. We can then formally define the concept of near-optimality as follows.

**Definition 5** ($\epsilon$-optimal neural network)**.** *A neural network with parameters $\theta$ is said to be $\epsilon$-optimal if there exists a small positive number $\delta$ such that $\tilde{Q}_\theta(s_1, \lambda) \geq \max\{Q_{\lambda,act}(s_1), Q_{\lambda,pass}(s_1)\} - \epsilon$ for all $s_0, s_1$, and $\lambda \in [f_\theta(s_0) - \delta, f_\theta(s_0) + \delta]$.*

Having outlined the necessary terms, we now move to establishing a sufficient condition for the $\gamma$-Whittle-accuracy of a neural network applied on $Env(\lambda)$.

**Theorem 1.** *If the arm is strongly indexable, then for any $\gamma > 0$, there exists a positive $\epsilon$ such that any $\epsilon$-optimal neural network controlling $Env(\lambda)$ is also $\gamma$-Whittle-accurate.*

*Proof.* For a given $\gamma > 0$, let,

$$\epsilon = \min_s\{\min\{Q_{W(s)+\gamma,pass}(s) - Q_{W(s)+\gamma,act}(s), Q_{W(s)-\gamma,act}(s) - Q_{W(s)-\gamma,pass}(s)\}\}/2.$$

Since the arm is strongly indexable and $W(s)$ is its Whittle index, we have $\epsilon > 0$.

We prove the theorem by establishing the following equivalent statement: If the neural network is not $\gamma$-Whittle-accurate, then there exists states $s_0, s_1$, activation cost $\lambda \in [f_\theta(s_0) - \delta, f_\theta(s_0) + \delta]$, such that the discounted net reward of applying a neural network to $Env(\lambda)$ with initial state $s_1$ is strictly less than $\max\{Q_{\lambda,act}(s_1), Q_{\lambda,pass}(s_1)\} - \epsilon$.

Suppose the neural network is not $\gamma$-Whittle-accurate, then there exists a state $s'$ such that $|f_\theta(s') - W(s')| > \gamma$. We set $s_0 = s_1 = s'$. For the case $f_\theta(s') > W(s') + \gamma$, we set $\lambda = f_\theta(s') + \delta$. Since $\lambda > W(s') + \gamma$, we have $\max\{Q_{\lambda,act}(s'), Q_{\lambda,pass}(s')\} = Q_{\lambda,pass}(s')$ and $Q_{\lambda,pass}(s') - Q_{\lambda,act}(s') \geq 2\epsilon$. On the other hand, since $f_\theta(s') > \lambda$, the neural network would activate the arm in the first round and its discounted reward is at most $Q_{\lambda,act}(s') < Q_{\lambda,pass}(s') - 2\epsilon < \max\{Q_{\lambda,act}(s'), Q_{\lambda,pass}(s')\} - \epsilon$.

For the case $f_\theta(s') < W(s') - \gamma$, a similar argument shows that the discounted reward for the neural network when $\lambda = f_\theta(s') - \delta$ is smaller than $\max\{Q_{\lambda,act}(s'), Q_{\lambda,pass}(s')\} - \epsilon$. This completes the proof.

□

## 4.2   Training Procedure for NeurWIN

Based on Thm. 1 and Def. 5, we define our objective function as $\sum_{s_0,s_1} \tilde{Q}_\theta(s_1, \lambda = f_\theta(s_0))$, with the estimated index $f_\theta(s_0)$ set as the environment's activation cost. A neural network that achieves a near-optimal $\sum_{s_0,s_1} \tilde{Q}_\theta(s_1, f_\theta(s_0))$ is also Whittle-accurate, which motivates the usage of deep reinforcement learning to find Whittle-accurate neural networks. Therefore we propose NeurWIN: an algorithm based on REINFORCE [31] to update $\theta$ through stochastic gradient ascent, where the

---
**Algorithm 1** NeurWIN Training
---
**Input: Parameters $\theta$, discount factor $\beta \in (0,1)$, learning rate $L$, sigmoid parameter $m$, mini-batch size $R$.**
**Output: Trained neural network parameters $\theta^+$.**
**for** each mini-batch $b$ **do**
    Choose two states $s_0$ and $s_1$ uniformly at random, and set $\lambda \leftarrow f_\theta(s_0)$ and $\bar{G}_b \leftarrow 0$
    **for** each episode $e$ in the mini-batch **do**
        Set the arm to initial state $s_1$, and set $h_e \leftarrow 0$
        **for** each round $t$ in the episode **do**
            Choose $a[t] = 1$ w.p. $\sigma_m(f_\theta(s[t]) - \lambda)$, and $a[t] = 0$ w.p. $1 - \sigma_m(f_\theta(s[t]) - \lambda)$
            **if** $a[t] = 1$ **then**
                $h_e \leftarrow h_e + \nabla_\theta \ln(\sigma_m(f_\theta(s[t]) - \lambda))$
            **else**
                $h_e \leftarrow h_e + \nabla_\theta \ln(1 - \sigma_m(f_\theta(s[t]) - \lambda))$
            **end if**
        **end for**
        $G_e \leftarrow$ empirical discounted net reward in episode $e$
        $\bar{G}_b \leftarrow \bar{G}_b + G_e/R$
    **end for**
    $L_b \leftarrow$ learning rate in mini-batch $b$
    Update parameters through gradient ascent $\theta \leftarrow \theta + L_b \sum_e (G_e - \bar{G}_b) h_e$
**end for**
---

gradient is defined as $\nabla_\theta \sum_{s_0, s_1} \tilde{Q}_\theta(s_1, f_\theta(s_0))$. For the gradient to exist, we require the output of the environment to be differentiable with respect to the input $f_\theta(s[t])$. To fulfill the requirement, we replace the step function in Fig. 1 with a sigmoid function,

$$\sigma_m(f_\theta(s[t]) - \lambda) := \frac{1}{1 + e^{-m(f_\theta(s[t]) - \lambda)}} \tag{1}$$

Where $m$ is a sensitivity parameter. The environment then chooses $a[t] = 1$ with probability $\sigma_m(f_\theta(s[t]) - \lambda)$, and $a[t] = 0$ with probability $1 - \sigma_m(f_\theta(s[t]) - \lambda)$. We call this differentiable environment $Env^*(\lambda)$.

The complete NeurWIN pseudocode is provided in Alg. 1. Our training procedure consists of multiple mini-batches, where each mini-batch is composed of $R$ episodes. At the beginning of each mini-batch, we randomly select two states $s_0$ and $s_1$. Motivated by the condition in Thm. 1, we consider the environment $Env^*(f_\theta(s_0))$ with initial state $s_1$, and aim to improve the empirical discounted net reward of applying the neural network to such an environment.

In each episode $e$ from the current mini-batch, we set $\lambda = f_\theta(s_0)$ and initial state to be $s_1$. We then apply the neural network with parameters $\theta$ to $Env^*(\lambda)$ and observe the sequences of actions $\big(a[1], a[2], \dots \big)$ and states $\big(s[1], s[2], \dots \big)$. We can use these sequences to calculate their gradients with respect to $\theta$ through backward propagation, which we denote by $h_e$. At the end of the mini-batch, NeurWIN would have stored the accumulated gradients for each of the $R$ mini-batch episodes to tune the parameters.

We also observe the discounted net reward and denote it by $G_e$. After all episodes in the mini-batch finish, we calculate the average of all $G_e$ as a bootstrapped baseline and denote it by $\bar{G}_b$. Finally, we do a weighted gradient ascent with the weight for episode $e$ being its offset net reward, $G_e - \bar{G}_b$.

When the step size is chosen appropriately, the neural network will be more likely to follow the sequences of actions of episodes with larger $G_e$ after the weighted gradient ascent, and thus will have a better empirical discounted net reward.

# 5 Experiments

## 5.1 Overview

In this section, we demonstrate NeurWIN's utility by evaluating it under three recently studied applications of restless bandit problems. In each application, we consider that there are $N$ arms and a controller can play $M$ of them in each round. We evaluate three different pairs of $(N, M)$: $(4, 1), (10, 1)$, and $(100, 25)$, and average the results of 50 independent runs when the problems are stochastic. Some applications consider that different arms can have different behaviors. For such scenarios, we consider that there are multiple types of arms and train a separate NeurWIN for each type. During testing, the controller calculates the index of each arm based on the arm's state and schedules the $M$ arms with the highest indices.

The performance of NeurWIN is compared against the proposed policies in the respective recent studies. In addition, we also evaluate the REINFORCE [31], Wolpertinger-DDPG (WOLP-DDPG) [10], Amortized Q-learning (AQL) [27], QWIC [11], and WIBQL [4]. REINFORCE is a classical policy-based RL algorithm. Both WOLP-DDPG and AQL are model-free deep RL algorithms meant to address problems with big action spaces. All three of them view a restless bandit problem as a Markov decision problem. Under this view, the number of states is exponential in $N$ and the number of possible actions is $\binom{N}{M}$, which can be as large as $\binom{100}{25} \approx 2.4 \times 10^{23}$ in our setting. Neither REINFORCE nor WOLP-DDPG can support such a big action space, so we only evaluate them for $(N, M) = (4, 1)$ and $(10, 1)$. On the other hand, QWIC and WIBQL aim to find the Whittle index through Q-learning. They are tabular RL methods and do not scale well as the state space increases. Thus, we only evaluate QWIC and WIBQL when the size of the state space is less than one thousand. We use open-source implementations for REINFORCE [12] and WOLP-DDPG [7].

In addition, we use experiment results to evaluate two important properties. First, we evaluate whether these three applications are strongly indexable. Second, we evaluate the performance of NeurWIN when the simulator does not perfectly capture the actual behavior of an arm.

We use the same neural network architecture for NeurWIN in all three applications. The neural network is a fully connected one that consists of one input layer, one output layer, and two hidden layers. There are 16 and 32 neurons in the two hidden layers. The output layer has one neuron, and the input layer size is the same as the dimension of the state of one single arm. As for the REINFORCE, WOLP-DDPG, AQL algorithms, we choose the neural network architectures so that the total number of parameters is slightly more than $N$ times as the number of parameters in NeurWIN to make a fair comparison. ReLU activation function is used for the two hidden layers. An initial learning rate $L = 0.001$ is set for all cases, with the Adam optimizer [15] employed for the gradient ascent step. The discount factor is $\beta = 0.99$ with an episode horizon of 300 timesteps. Each mini-batch consists of five episodes. More details can be found in the appendix.

## 5.2 Deadline Scheduling

A recent study [34] proposes a deadline scheduling problem for the scheduling of electrical vehicle charging stations. In this problem, a charging station has $N$ charging spots and enough power to charge $M$ vehicles in each round. When a charging spot is available, a new vehicle may join the system and occupy the spot. Upon occupying the spot, the vehicle announces the time that it will leave the station and the amount of electricity that it needs to be charged. The charging station obtains a reward of $1 - c$ for each unit of electricity that it provides to a vehicle.

However, if the station cannot fully charge the vehicle by the time it leaves, then the station needs to pay a penalty of $F(B)$, where $B$ is the amount of unfulfilled charge. The goal of the station is to maximize its net reward, defined as the difference between the amount of reward and the amount of penalty. In this problem, each charging spot is an arm. [34] has shown that this problem is indexable. We further show in Appendix B that the problem is also strongly indexable.

We use exactly the same setting as in the recent study [34] for our experiment. In this problem, the state of an arm is denoted by a pair of integers $(D, B)$ with $B \leq 9$ and $D \leq 12$. The size of state space is 120 for each arm.

The experiment results are shown in Fig. 2. It can be observed that the performance of NeurWIN converges to that of the deadline Whittle index in 600 training episodes. In contrast, other MDP

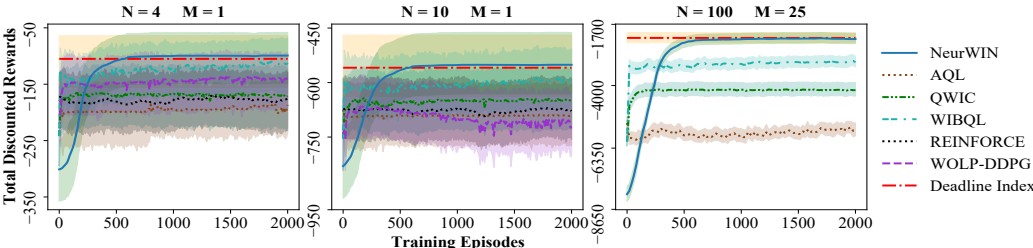

Figure 2: Average rewards and confidence bounds of different policies for deadline scheduling.

algorithms have virtually no improvement over 2,000 training episodes and remain far worse than NeurWIN. This may be due to the explosion of state space. Even when $N$ is only 4, the total number of possible states is $120^4 \approx 2 \times 10^8$, making it difficult for the compared deep RL algorithms to learn the optimal control in just $2,000$ episodes. QWIC performs poorly compared to NeurWIN and the deadline index, while WIBQL's performance degrades with more arms. The result suggest that both QWIC and WIBQL do not learn an accurate approximation of the Whittle index.

## 5.3 Recovering Bandits

The recovering bandits [21] aim to model the time-varying behaviors of consumers. In particular, it considers that a consumer who has just bought a certain product, say, a television, would be much less interested in advertisements of the same product in the near future. However, the consumer's interest in these advertisements may recover over time. Thus, the recovering bandit models the reward of playing an arm, i.e., displaying an advertisement, by a function $f(\min\{z, z_{max}\})$, where $z$ is the time since the arm was last played and $z_{max}$ is a constant specified by the arm. There is no known Whittle index or optimal control policy for this problem.

The recent study [21] considers the special case of $M = 1$. When the function $f(\cdot)$ is known, it proposes a $d$-step lookahead oracle. Once every $d$ rounds, the $d$-step lookahead oracle builds a $d$-step lookahead tree. Each leaf of the $d$-step lookahead tree corresponds to a sequence of $d$ actions. The $d$-step lookahead oracle then picks the leaf with the best reward and use the corresponding actions in the next $d$ rounds. As the size of the tree grows exponentially with $d$, it is computationally infeasible to evaluate the $d$-step lookahead oracle when $d$ is large. We modify a heuristic introduced in [21] to greedily construct a 20-step lookahead tree with $2^{20}$ leaves and pick the best leaf. [21] also proposes two online algorithms, RGP-UCB and RGP-TS, for exploring and exploiting $f(\cdot)$ when it is not known a priori. We incorporate [21]'s open-source implementations of these two algorithms in our experiments.

In our experiment, we consider different arms have different functions $f(\cdot)$. The state of each arm is its value of $\min\{z, z_{max}\}$ and we set $z_{max} = 20$ for all arms.

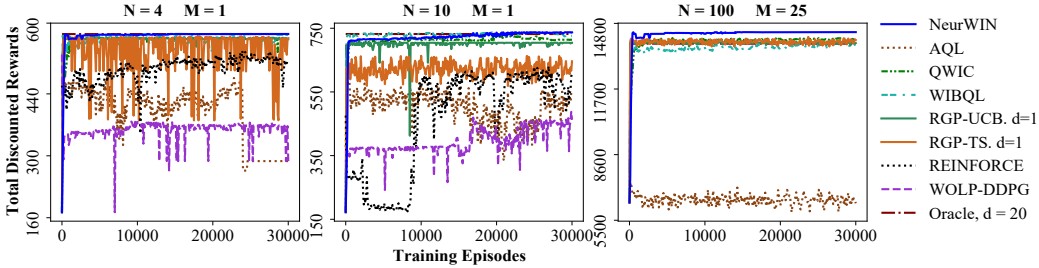

Figure 3: Experiment results for the recovering bandits.

Experiment results are shown in Fig. 3. It can be observed that NeurWIN is able to outperform the 20-step lookahead oracle in its respective setting with just a few thousands of training episodes. Other algorithms perform poorly.

## 5.4 Wireless Scheduling

A recent paper [1] studies the problem of wireless scheduling over fading channels. In this problem, each arm corresponds to a wireless client. Each wireless client has some data to be transmitted and it suffers from a holding cost of 1 unit per round until it has finished transmitting all its data. The channel quality of a wireless client, which determines the amount of data can be transmitted if the wireless client is scheduled, changes over time. The goal is to minimize the sum of holding costs of all wireless clients. Equivalently, we view the reward of the system as the negative of the total holding cost. Finding the Whittle index through theoretical analysis is difficult. Even for the simplified case when the channel quality is i.i.d. over time and can only be in one of two possible states, the recent paper [1] can only derive the Whittle index under some approximations. It then proposes a *size-aware index* policy using its approximated index.

In the experiment, we adopt the settings of channel qualities of the recent paper. The channel of a wireless client can be in either a good state or a bad state. Initially, the amount of load is uniform between 0 and 1Mb. The state of each arm is its channel state and the amount of remaining load. The size of state space is $2 \times 10^6$ for each arm. We consider that there are two types of arms, and different types of arms have different probabilities of being in the good state. During testing, there are $\frac{N}{2}$ arms of each type.

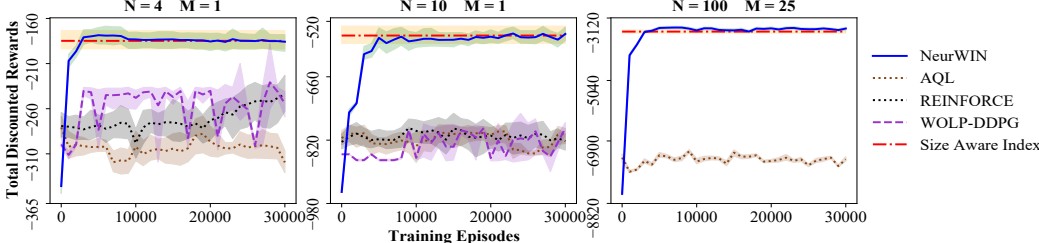

Figure 4: Average rewards and confidence bounds of different policies for wireless scheduling.

Experiment results are shown in Fig. 4. It can be observed that NeurWIN is able to perform as well as the size-aware index policy with about $5,000$ training episodes. It can also be observed that other learning algorithms perform poorly.

## 5.5 Evaluation of NeurWIN's Limitations

A limitation of NeurWIN is that it is designed for strongly indexable bandits. Hence, it is important to evaluate whether the considered bandit problem is strongly indexable. Recall that a bandit arm is strongly indexable if $D_s(\lambda)$ is strictly decreasing in $\lambda$, for all states $s$. We extensively evaluate the function $D_s(\lambda)$ of different states for the three applications considered in this paper. We find that all of them are strictly decreasing in $\lambda$. Fig. 5 shows the function $D_s(\lambda)$ of five randomly selected states for each of the three applications. The deadline and wireless scheduling cases were averaged over 50 runs. These results confirm that the three considered restless bandit problems are indeed strongly indexable.

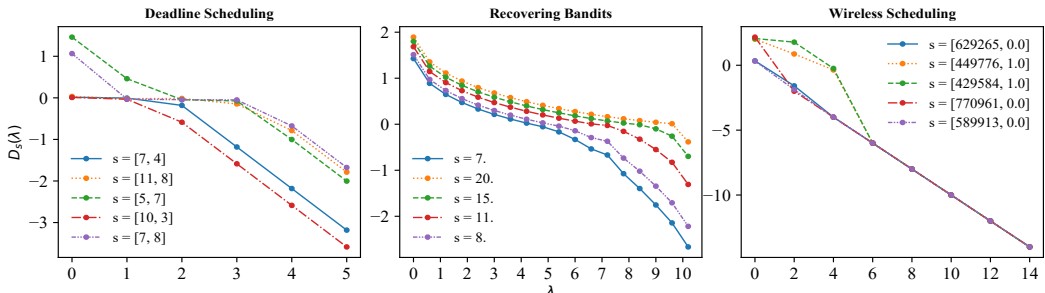

Figure 5: Difference in discounted net reward $D_s(\lambda)$ for randomly selected initial states.

Another limitation of NeurWIN is that it requires a simulator for each arm. To evaluate the robustness of NeurWIN, we test the performance of NeurWIN when the simulator is not perfectly precise. In particular, let $R_{act}(s)$ and $R_{pass}(s)$ be the rewards of an arm in state $s$ when it is activated and not activated, respectively. Then, the simulator estimates that the rewards are $R'_{act}(s) = (1 + G_{act,s})R_{act}(s)$ and $R'_{pass}(s) = (1 + G_{pass,s})R_{pass}(s)$, respectively, where $G_{act,s}$ and $G_{pass,s}$ are independent Gaussian random variables. The variance of these Gaussian random variables correspond to the magnitude of root mean square errors in the simulator.

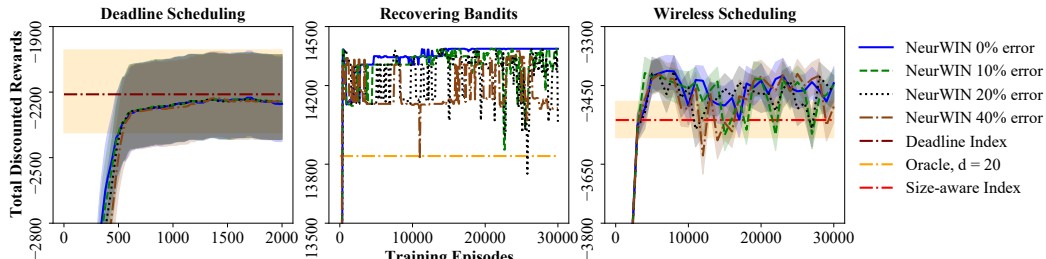

Figure 6: Experiment results for NeurWIN with noisy simulators.

We train NeurWIN using the noisy simulators with different levels of errors for the three problems. For each problem, we compare the performance of NeurWIN against the respective baseline policy. Unlike NeurWIN, the baseline policies make decisions based on the true reward functions rather than the estimated ones. The results for the case $N = 100$ and $M = 25$ are shown in Fig. 6. It can be observed that the performance of NeurWIN only degrades a little even when the root mean square error is as large as 40% of the actual rewards, and its performance remains similar or even superior to that of the baseline policies.

## 6 Conclusion

This paper introduced NeurWIN: a deep RL method for estimating the Whittle index for restless bandit problems. The performance of NeurWIN is evaluated by three different restless bandit problems. In each of them, NeurWIN outperforms or matches state-of-the-art control policies. The concept of strong indexability for bandit problems was also introduced. In addition, all three considered restless bandit problems were empirically shown to be strongly indexable.

There are several promising research directions to take NeurWIN into: extending NeurWIN into the offline policy case. One way is to utilize the data samples collected offline to construct a predictive model for each arm. Recent similar attempts have been made for general MDPs in [14, 33]. NeurWIN would then learn the Whittle index based on this predictive model of a single arm, which is expected to require fewer data samples due to the decomposition provided by index policies.

Another direction is to investigate NeurWIN's performance in cases with non-strong indexability guarantees. For cases with verifiably non strongly-indexable states, one would add a pre-processing step that provides performance precision thresholds on non-indexable arms.

## Acknowledgment

This material is based upon work supported in part by the U.S. Army Research Laboratory and the U.S. Army Research Office under Grant Numbers W911NF-18-1-0331, W911NF-19-1-0367 and W911NF-19-2-0243, in part by Office of Naval Research under Contracts N00014-18-1-2048 and N00014-21-1-2385, in part by the National Science Foundation under Grant Numbers CNS 1955696 and CPS-2038963, and in part by the Ministry of Science and Technology of Taiwan under Contract Numbers MOST 109-2636-E-009-012 and MOST 110-2628-E-A49-014.

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
