# OpenReview forum: "NeurWIN: Neural Whittle Index Network For Restless Bandits Via Deep RL"
_NeurIPS.cc/2021/Conference — NeurIPS 2021 Poster_

### Official Review · Reviewer_WCWa · 2021-07-01

**Rating:** 6
**Confidence:** 5

**Summary:**


Whittle index are used to construct a powerful heuristics for restless bandit problem.  This paper proposes to estimate Whittle Index in discounted Restless bandit problem via a deep reinforcement learning algorithm. The authors define a notion of strong indexability that they use to construct a deep RL algorithm that learns Whittle indices. The authors argue that when a problem is strongly indexability and the neural network is precise enough, the deep RL algorithm will learn Whittle index (although their theoretical result is not fully convincing to me).

The authors present extensive experimental results that show that their algorithm compute Whittle indices and provide a very good performance compared to existing work.


**Ethical Concerns:**

NA.

**Limitations And Societal Impact:**

Could the authors explain better what is the implication of Theorem 3? Does it guarantee anything on the convergence of the algorithm?

The pseudo-code of the algorithm is very hard to understand. Could the authors justify their design choices and the potential impact of them?

Is there an open-source implementation of the code that would make it usable by others?

**Main Review:**

I liked the paper and the algorithm it proposes. While Whittle indices have been defined a long time ago, there is a recent surge of interest on restless bandits since it has potential application in many RL problems. I liked the experimental section that I find quite thorough and convincing. The comparison with related work is interesting as well as the fact that the authors show the strong indexability of various problems (either theoretically for the first, or numerically). This indicates that this notion is adapted. The algorithm performs very well in three real-life problems. Also, the limitation of the algorithm is provided: the authors test the case where simulator is noisy and the strong indexability of the three problems. All-in-all, this convinced me that the presented algorithm has potential application for many restless bandit learning algorithms.

That being said, I think that the novelty of the paper is somehow limited and I think that some part would deserve to be better explained. In particular, I did not find Theorem 3 very informative: I found the current version of the theorem rather weak and it does not convince me that a rich-enough network will indeed learn the true Whittle indices.

To me, it is not clear how others algorithms are trained (the same number of episodes and batches as NeurWIN?). And about others algorithms structures (almost the same number of parameters as NeurWIN, but how the hidden layers, input and output layers are connected?).

The NeurWIN algorithm seems to need an internal simulator. It is an off-policy algorithm. Can we say something about its regret (at least empirically?) or in general about its performance while learning?

Also, Whittle indices are particularly useful for time-average reward and not for the discounted case. Would it still work here?

Minor remark: In the organization of the paper, I do not understand why section 3.3 is where it is. This section is interesting but completly breaks the flow of the paper. To me, this should be moved to section 5 (or merged with section 5.5 in a new section).


** Rebuttal **

The answer of the authors clarified my view of the paper. I am raising my score from 5 to 6 (not higher because I am still puzzled about the novelty of the approach, and because incorporating the new theory in a convincing manner might not be that trivial.)

**Time Spent Reviewing:**

3

---

> ### Author Response · Authors · 2021-08-10
> **Response (1/1)**
>
> We thank the reviewer for their helpful comments and suggestions.
>
>
> ### Theorem 3 comment: "I did not find Theorem 3 very informative..."
> We provide a better explanation for theorem 3 and required definitions as requested by all reviewers,
>
> Consider a neural network with parameter $\theta$. Let $ \tilde{Q_\theta}(s, \lambda)$ be the average reward of applying this neural network to Env($\lambda$) with initial state $s$. If the neural network is indeed the optimal controller to Env($\lambda$), then we have $\tilde{Q_\theta}(s, \lambda)= \max\\{Q_{\lambda, act}(s), Q_{\lambda, pass}(s)\\}$, for all $s$ and $\lambda$. We further define the near-optimality of the neural network as follows:
>
> Definition: A neural network with parameter $\theta$ is said to be $\epsilon$-optimal if there exists a positive number $\delta$ such that $\tilde{Q_\theta}(s_1, \lambda)\geq \max\\{Q_{\lambda, act}(s_1), Q_{\lambda, pass}(s_1)\\}-\epsilon$, for all $s_0, s_1$ and $\lambda\in [f_\theta(s_0)-\delta, f_{\theta}(s_0)+\delta]$.
>
> We then show that a near-optimal neural network is also Whittle-accurate.
>
> Revised Theorem 3: If the arm is strongly indexable, then for any $\gamma>0$, there exists a positive $\epsilon$ such that every $\epsilon$-optimal neural network is also $\gamma$-Whittle-accurate.
>
> Based on the revised Theorem 3, our goal is to train a near-optimal neural network. We define our objective function as $\sum_{s_0, s_1} \tilde{Q}_\theta(s_1, f_\theta(s_0))$. In each iteration, we randomly choose $s_0, s_1$ and aim to update $\theta$ based on $\nabla_\theta \tilde{Q}_\theta(s_1, f_\theta(s_0))$. To this end, we employ the REINFORCE algorithm as described in Alg. 1.
>
> ---
>
> ### "To me , it is not clear how other algorithms are trained..."
>
> We offer detailed setting description in appendix D,E,F for the three cases, which can be found in the supplementary material.
> All benchmark algorithms trained a network with two hidden fully connected layers (same as NeurWIN).
> With NeurWIN is trained on a single arm, the MDP algorithms are trained on a set N of arms that form the MDP. The MDP state $S[t]$ at time $t$ is defined by the arms' states $S[t] = (s_1[t], s_2[t], ... , s_N[t])$.
> For N = 4, MDP algorithms networks would have slightly larger parameter count than 4*(single NeurWIN network number of parameters).
> The environment trains the algorithms on episode batches where the first state is the same across batches. Total timesteps are the same for MDP algorithms and NeurWIN.
>
> ---
>
> ### "Whittle indices are particularly useful for time-average reward... Would it still work here?"
> We trained NeurWIN for $\beta$ = 1 in the deadline and recovering bandits cases. The wireless scheduling setting is not applicable to the time-average reward setting, since with $\beta$ = 1, arms would have an infinite total holding cost, and the optimal policy becomes not activating the arm (optimal policy would be to leave the arm passive for all timesteps).
> Results show NeurWIN is able to operate in the average-reward setting, and achieve similar or better performance in deadline scheduling and recovering bandits, respectively. Numerical results:
>
> **deadline scheduling with $\beta$ = 1**
>
> deadline whittle index N = 4  M =1 : -329.904
>
> deadline whittle index N = 10 M =1 : -1782.384
>
> deadline whittle index N = 100  M =25 : -6984.072
>
> |Policy | episode 0  | episode 500  | episode 1000  | episode 1500  | episode 2000  |
> | ------------- |-------------| -------------| -------------|-------------|-------------|
> | N = 4 M =1   |-962.204 |-336.280 | -314.526| -313.522 | -312.212 |
> | N = 10 M =1    | -2650.140 |-1789.336 | -1758.088 | -1759.024| -1758.460 |
> | N = 100 M = 25 | -25714.908| -7279.484 | -7024.088 | -7000.196 | -7002.824|
>
>
>
> **recovering bandits with $\beta$ = 1**
>
> 8-lookahead for N = 4  M = 1 : 1813.53
>
> 20-lookahead for N = 4  M = 1 : 1826.88
>
> 8-lookahead for N = 10  M = 1 : 2324.89
>
> 20-lookahead for N = 10  M = 1 : 2335.27
>
> 8-lookahead for N = 100  M = 25 : 43291.20
>
> 20-lookahead for N = 100  M = 25 : 43509.00
>
>
> |Policy | episode 0  | episode 5000  | episode 10000  | episode 15000  | episode 20000  | episode 25000  | episode 30000  |
> | ------------- |-------------| -------------| -------------|-------------|-------------|-------------|-------------|
> | N = 4 M = 1 |   543.808  | 1820.599 | 1824.550 | 1826.877 | 1826.877| 1826.877| 1826.877 |
> | N = 10 M = 1  |  543.808   |2285.232 | 2285.266 |2329.576 |2354.366 |  2353.447| 2352.043 |
> | N = 100 M = 25   |   19987.374   | 45514.965 | 45613.750| 45671.930 | 45671.930 | 45671.930 | 45671.930 |
>
> ---
>
> ### "I do not understand why section 3.3 is where it is."
> We will move section 3.3 to the appendices, and incorporate the new explanation of theorem 3, along with more expansive algorithm description, in the updated paper.
>
> ---
>
> ### "Is there an open-source implementation of the code that would make it usable by others?"
>
> We have already provided the source code in the supplementary material along with the appendices.  We will add a note in the main paper so that it is clear where to find the code.  We will also provide a public code repository once the discussion period ends.

---

### Official Review · Reviewer_jp8H · 2021-07-16

**Rating:** 6
**Confidence:** 4

**Summary:**

The authors develop a reinforcement learning based method for computing the Whittle Index heuristic policy for solving Restless Bandit problems. This is a novel methodological contribution in the space of restless bandits. Several experimental results are provided demonstrating the good performance and general applicability of the method.

**Limitations And Societal Impact:**

yes

**Main Review:**

This paper presents an original algorithm that will be of general interest to researchers in the restless bandits space. Specifically, the authors seem to be the first to use neural-network-based function approximation for learning the Whittle index policy. The paper is relatively well structured and written, though some key concepts are unclear (more comments provided below). The experimental results are comprehensive and the domains tested are strong ones, making their results fairly convincing, with a few minor comments. However, authors also missed a key competitor in related work. I think this work has potential to be a good submission, but I ask for the following clarities.

Major comments:
1. Thm 3 is reasonable in terms of motivating the use of reinforcement learning for finding the Whittle index. However, I feel like what is missing from this paper is a Thm 4 that proves that algorithm 1 is valid/will converge to the proper values of the Whittle index -- I expound on this in the next comment.
I am having some trouble determining the validity of algorithm 1. Perhaps its presentation could be improved. As far as I can tell, a key component of the validity of the algorithm is that the \lambda value is set by the network at the start of each minibatch of training, which itself sets the cost of acting in the environment. If we assume that this method converges, then eventually we would expect that that lambda value would be set to the Whittle index, i.e., the value that makes the planner expect to get exactly 0 reward for following the optimal policy -- and since you are actually charging that \lambda value in the empirical rewards, it seems that the trajectories would also go to 0 in expectation. However, an alternative controller which sets \lambda to 0 and always predicts 0 would always act randomly, but never get charged any cost of acting -- why would a neural network not find the second option preferable if it receives greater reward? This does not seem to be what happens, since the experimental results are convincing. So it would greatly help to have both an intuitive, and really a theoretical justification for why the authors' algorithm converges.

2. Line 251 - 253: "For such scenarios, we consider that there are multiple types of arms and train a separate NeurWIN for each type"
 - it would be nice to see also an experiment where every arm is sampled randomly from some distribution or set of distributions, since the assumption that many arms are exactly the same seems strong and could lead to cases where the Whittle indexes that need to be learned for different arm types have significant numerical separation that makes the precision of the authors' method less important for achieving good performance in the environment. This may increase computational demand, but seems reasonable to run so that we can see experimentally that the authors' method is learning the Whittle indices to a non-trivial precision.

3. Authors missed an important competitor which proposes a Q-learning algorithm for learning the Whittle index which comes with convergence guarantees (bibtex below). Authors implement and compare against this baseline, as it seems to be the closest competitor to their algorithm. However, since the authors' method is based on deep Q-learning, it is more general than the below competitor that uses tabular Q-learning. However, there is likely overlap in the theory regarding the proof of convergence of the competitor method, and the algorithmic ideas behind NeurWIN that should be elucidated by the authors to properly contextualize the authors' contribution.

@article{avrachenkov2020whittle,
  title={Whittle index based Q-learning for restless bandits with average reward},
  author={Avrachenkov, Konstantin and Borkar, Vivek S},
  journal={arXiv preprint arXiv:2004.14427},
  year={2020}
}

Medium Comments:
4. Fig 3 - how well does 20-step lookahead perform on (4,1) and (10,1)? Also why does QWIC outperform the lookahead on (4,1) but not (10,1)?

5. The authors of [17] implement UCB and Thompson-Sampling based algorithms -- it would be helpful to see how NeurWIN compares to these baselines on at least the recovering bandits environment.

6. Line 338- 340: "Fig. 5 shows the function Ds(λ) of five randomly selected states for each of the three applications. This result suggests that most practical bandit problems are indeed strongly indexable.”
"Most" is far too strong a claim based off the N=3 experimental domains given as evidence. This evidence is at best a proof of concept. A far more principled investigation of what it means to be (and not to be) strongly indexable in terms of problem structure would be necessary here to a claim about "most" bandits and more context would need to be given about what is meant by "most", e.g., most "real-world bandits?", "most bandits in literature?", "most out of all possible bandits?", etc.

7. Throughout the paper, the authors make implicit claims that most problems are indexable without giving much support for that claim. Perhaps because the authors' algorithm relies on problems being indexable in order to be asymptotically optimal. Some things that could help strengthen this paper in that regard:
  a) Adapt/reframe the numerical "strong indexability" verification step that authors' use in experiments to be a pre-processing step to their algorithm. This way, users can at least numerically verify to their desired level of precision whether or not the authors' algorithm will be optimal for their use-case.
  b) Argue for the strength of the Whittle index approach in non-indexable or non-verifiably-indexable (e.g., very large state space) cases by pointing to the  provided experimental results or pointing to papers that have analyzed/experimented with the Whittle index policy in such cases.

Minor Comments:
line 32- 33: "where the index loosely corresponds to the amount of cost that we are willing to pay to play the arm"
- Authors should consider rewording this. The Whittle index is defined as exactly this cost that the planner is willing to pay to play the arm, so I am not certain that saying "loosely corresponds" is accurate.

33-34: "It has been shown that the Whittle index policy is either optimal or asymptotically optimal in many settings."
- Probably makes sense to introduce indexability here? E.g., “asymptotically optimal under a technical condition known as indexability”. In general it would be helpful to frame the story of RMABs in the light of how the problem has typically been solved, i.e., by proving or providing evidence of indexability, then deriving Whittle index policies. Then making clear how your algorithm will improve on this style of approach and why it is valid.

35-36: "In this paper, we present Neural Whittle Index Network (NeurWIN), a principled machine learning approach that finds the Whittle indices for virtually all restless bandit problems."
- The Whittle index does not always exist, i.e., when a problem is not indexable. So this statement is somewhat misleading and should be explained more carefully. Can you be more precise by what you mean by "virtually all restless bandit problems"?

Line 66: "There has been a lot of studies"
 - Just suggesting something slightly more formal thant "a lot", e.g., "There have been many studies"

In definition 2, I was mildly confused why s was in A(\lambda)... but realized that A(\lambda) is indeed defined as a set of states. I would lightly suggest using something besides A for a set of states (perhaps \sigma?), since lower-case a is used for actions.

line 142: "attractable" --> attractive

line 142-143: "Intuitively, as the activation cost increases, it becomes less attractable to activate the arm in any given state. Hence, one would expect Ds(λ) to be strictly decreasing in λ for most practical problems."
 - While I tend to agree with this claim, I think the authors should make an effort to support this argument that "most problems are strongly indexable" with evidence (perhaps mention the experimental results) more so than intuition or by arguing further about the typical structure of problems. While it seems reasonable that increasing activation cost would not increase attractiveness, it seems perfectly plausible that increasing activation cost would leave attractiveness unchanged in some problems with specific structure.

Thm 1. -- this is mostly a restatement of existing concepts, i.e., indexability and the \lambda-adjusted Q-values, so I would consider reassigning this as a proposition or something similar. To this reviewer, a theorem implies a novel theoretical contribution.

Thm 2. -- It is unclear to me why strong indexability is important. It is not a concept I am familiar with from other restless bandit literature and unclear how it helps the authors' algorithm. Strong indexability seems to guarantee that there should be one unique \lambda that makes D(s,\lambda)=0, but again, it would help if the authors could clarify why such a concept is useful. For instance, such a concept is not required by Weber and Weiss in the proof of the optimality of the Whittle index. Understanding strong indexability is important to this reviewer, since the authors present these results as Theorems in their paper. Perhaps it is required for their neural network system to be differentiable everywhere?

- Recovering bandits is a strong experimental domain for the authors' algorithm, since the indexability of this space is unknown. It would help to highlight explicitly this fact in the introduction to make clear the algorithm's general applicability.
---------------------------------------------
Update:

I have read the other reviews and the author responses. I thank the authors for their response. Overall I am now convinced about the empirical results, but still unsure about the theory and the algorithm itself.

Experiments: The authors thoroughly addressed all my comments about experimental results, including implementations of WIBQL and UCB/Thompson Sampling from [17] and the results seem convincing enough, though I agree with Ha51 that WIBQL requires much hyperparameter tuning to be most effective, so it would be good to have more details there.

Theory/Algorithm: I am also still not sure how well the theory actually supports the algorithm, even with the revised theorems. I am also still somewhat lacking in my understanding of why exactly the proposed algorithm should converge, and unfortunately, the authors did not address that concern in their response (at least not clearly).

Ultimately, I am increasing my score by 1 to reflect the authors addressing my empirical concerns. However, authors should make their code publicly available upon acceptance to facilitate others building on the authors' contributions despite some lack of clarity in the algorithmic descriptions.

**Time Spent Reviewing:**

5

---

> ### Author Response · Authors · 2021-08-10
> **Response (1/2)**
>
> We thank the reviewer for the constructive criticism.
>
> ### Major comment 1: "Thm3 is reasonable in terms of motivating the use of reinforcement learning... I feel like what is missing from this paper is a Thm 4"
>
> We will revise the statement of  Thm 3 in the paper based on the description below,
>
>
> Consider a neural network with parameter $\theta$. Let $ \tilde{Q_\theta}(s, \lambda)$ be the average reward of applying this neural network to Env($\lambda$) with initial state $s$. If the neural network is indeed the optimal controller to Env($\lambda$), then we have $\tilde{Q_\theta}(s, \lambda)= \max\\{Q_{\lambda, act}(s), Q_{\lambda, pass}(s)\\}$, for all $s$ and $\lambda$. We further define the near-optimality of the neural network as follows:
>
> Definition: A neural network with parameter $\theta$ is said to be $\epsilon$-optimal if there exists a positive number $\delta$ such that $\tilde{Q_\theta}(s_1, \lambda)\geq \max\\{Q_{\lambda, act}(s_1), Q_{\lambda, pass}(s_1)\\}-\epsilon$, for all $s_0, s_1$ and $\lambda\in [f_\theta(s_0)-\delta, f_{\theta}(s_0)+\delta]$.
>
> We then show that a near-optimal neural network is also Whittle-accurate.
>
> Revised Theorem 3: If the arm is strongly indexable, then for any $\gamma>0$, there exists a positive $\epsilon$ such that every $\epsilon$-optimal neural network is also $\gamma$-Whittle-accurate.
>
> Based on the revised Theorem 3, our goal is to train a near-optimal neural network. We define our objective function as $\sum_{s_0, s_1} \tilde{Q}_\theta(s_1, f_\theta(s_0))$. In each iteration, we randomly choose $s_0, s_1$ and aim to update $\theta$ based on $\nabla_\theta \tilde{Q}_\theta(s_1, f_\theta(s_0))$. To this end, we employ the REINFORCE algorithm as described in Alg. 1.
>
>
> Also, for the comment “If we assume that this method converges, then eventually we would expect that that lambda value would be set to the Whittle index….” We want to point out that, after convergence, the lambda value is the Whittle index for the state $s_0$, but not $s_1$. While the optimal policy would get 0 net reward if the initial state is $s_0$, it will not get 0 net reward if the initial state is $s_1$.
>
> ---
>
> ### Major comment 2: "it would be nice to see also an experiment where every arm is sampled randomly from some distribution ..."
> We have trained NeurWIN on randomly sampled arms for the three cases from the ranges:
> 1) deadline scheduling: penalty function $F_i(x)$ random parameters for an arm $i$ are $F_i(x) = a_i \cdot x^{b_i}$, with  $0 < a_i < 1$, and $1 < b_i < 3 $.
>
> 2) recovering bandits: random recovering reward function parameters for an arm $i$:   5 < $\theta_{0,i}$ < 10,  0.1 < $\theta_{1,i}$ < 0.9
>
> 3) wireless scheduling: random good channel probability for arms randomly selected between 0 and 1.
>
> Other parameters are kept the same as in the original case described in section 5.1 ($\beta$ = 0.99, episode horizon = 300 timesteps).
> We observe that NeurWIN is able to outperform baselines in the recovering bandits' case, and offer similar performance in the deadline and wireless scheduling cases in terms of total discounted rewards. In the case of deadline scheduling, we experience slight degradation in convergence to the true Whittle index compared with the original setting for N = 4 and N = 100. N = 4 needs 1220 episodes to match the deadline Whittle index total discounted rewards; N = 10 needs 1450 episodes, and 800 episodes are needed for N = 100 to reach the same Whittle index rewards' performance.
>
> For wireless scheduling case, NeurWIN converges to the size aware index using more episodes compared to the original setting. NeurWIN learned policy achieves the same size aware performance in 5800 episodes for N = 4, 8000 episodes for N = 10, and 4900 episodes for N = 100.
> As requested, we included the 20-lookahead result in recovering bandits for N = 4 in addition to 8-lookahead.
> Results show NeurWIN's robustness in learning the control policy of randomly sampled bandits.
> Numerical results for N = 4 M = 1 (N = 10 and N = 100 results are available upon request):
>
> **deadline scheduling  N = 4 M = 1**
>
> |Policy | episode 0  | episode 500  | episode 1000  | episode 1500  | episode 2000  |
> | ------------- |-------------| -------------| -------------|-------------|-------------|
> | NeurWIN  | -1497.526| -660.173| -556.307| -550.158|-569.577 |
> | Deadline Whittle Index  | -526.069| -526.069| -526.069| -526.069| -526.069|
> | REINFORCE   | -973.471 | -840.156 | -821.836 | -819.332 | -809.838 |
> | WOLP-DDPG | -863.638 |  -699.723 | -688.017 | -656.364 | -643.084 |
> | AQL   |  -960.081 | -1098.023 | -1114.300 |  -766.333 |  -749.505 |
> | QWIC  | -1340.586 | -879.996 |  -845.465 | -931.739 |  -880.671 |
> | WIBQL   | -1340.586 |  -658.252 |  -693.306 |  -673.889 |  -616.699 |
>
>
> **recovering bandits  N = 4 M = 1**
>
> |Policy | episode 0  | episode 5000  | episode 10000  | episode 15000  | episode 20000  | episode 25000  | episode 30000  |
> | ------------- |-------------| -------------| -------------|-------------|-------------|-------------|-------------|
> | NeurWIN | 92.692 | 479.309| 480.385 | 480.385 | 480.385| 478.635| 480.385|
> |8-lookahead |  478.597   | 478.597 | 478.597 | 478.597 | 478.597 | 478.597 | 478.597 |
> | 20-lookahead  | 480.854  | 480.854 | 480.854 | 480.854 | 480.854 | 480.854 | 480.854 |
> | REINFORCE  | 373.405 | 400.946 | 419.455 | 431.748 | 436.049 | 438.551 | 429.268 |
> | WOLP-DDPG | 454.786 | 376.174 | 412.617 | 438.238 | 452.377 | 449.813 | 457.547 |
> | AQL  | 378.713 | 376.183 | 334.807 | 334.329 | 368.402 | 377.420 | 355.478 |
> | QWIC  | 92.691 | 470.004 | 473.374 | 478.051 | 430.398 | 480.854 | 480.854 |
> | WIBQL | 92.692 | 480.787 | 466.614 | 479.960 | 480.488 | 480.787 | 480.787 |
>
>
> **Wireless scheduling  N = 4 M = 1**
>
> |Policy | episode 0  | episode 5000  | episode 10000  | episode 15000  | episode 20000  | episode 25000  | episode 30000  |
> | ------------- |-------------| -------------| -------------|-------------|-------------|-------------|-------------|
> | NeurWIN |  -335.345   | -190.579| -174.938 |  -170.484 | -170.788 | -171.004 | -170.626 |
> | size-aware index |  -175.191   | -175.191| -175.191| -175.191|-175.191 | -175.191| -175.191|
> | REINFORCE  |  -272.356   | -269.920 | -270.279 | -254.135 |-257.627| -247.590| -247.061|
> | WOLP-DDPG  |  -309.459 | -309.459 | -304.543 | -281.185 | -272.322 | -261.598  | -268.650 |
> | AQL  |  -272.574 | -309.471 | -306.708 | -310.689 | -310.356 | -307.945|-309.165 |

---

> > ### Author Response · Authors · 2021-08-10
> > **Response (2/2)**
> >
> >
> > ### Major comment 3: "Authors missed an important competitor which proposes a Q-learning algorithm for learning the Whittle index ..."
> >
> > We thank the reviewer for the comment. We have implemented the policy in the paper (referred to as WIBQL in the table). The results show that our NeurWIN scales better with larger state spaces. We are able to achieve better performance because of two important differences: First, as the reviewer pointed out, the Avrachenkov paper is a tabular method, which as shown it scales poorly with larger state spaces (e.g. deadline scheduling). Second, the Avrachenkov paper needs to learn the Q-value first, and then use the learned Q-value to adjust the Whittle index. In contrast, NeurWIN learns the Whittle index directly. These two differences also lead to very different algorithm designs. As explained in #1, our algorithm is effectively doing stochastic gradient descent with respect to the objective function $\sum_{s_0, s_1} \tilde{Q}_\theta(s_1, f_\theta(s_0))$.
> >
> > We also note that we tried running WIBQL for the wireless scheduling case, but the tabular method failed to run for a state space size per arm of 2 million states.
> > From the results, WIBQL performs well for small state spaces (recovering bandits with 20 states per arm), performs worse compared to NeurWIN in deadline scheduling (120 states per arm), and fails to run for the wireless scheduling case (2 million states per arm).
> > We will include the experiment results and the above discussions in the paper. Numerical results for WIBQL and NeurWIN:
> >
> > **Deadline scheduling**
> >
> > |Policy | episode 0 | episode 200 | episode 400   | episode 500  | episode 800  | episode 1000 | episode 1500  | episode 2000  |
> > | ------------- |-------------| -------------| -------------|-------------|-------------| -------------|-------------|-------------|
> > | NeurWIN  N = 4 M = 1 |  -301.034   |  -181.607  |  -116.450 |  -108.173 |  -100.366  | -99.768  | -99.472 |-99.250 |
> > | WIBQL   N = 4 M = 1  |   -241.982  | -153.848  | -130.753  | -121.844 |  -118.412 | -126.747   | -115.395 |-112.080 |
> > | NeurWIN  N = 10 M = 1 |  -829.923  | -675.260  |  -580.247  | -566.072  | -553.248 |  -551.771  | -551.325 |-551.100 |
> > | WIBQL   N = 10 M = 1  |  -753.276   | -624.944  |   -613.419  | -609.431 |  -616.215  | -607.776 |  -598.347 |-592.125 |
> > | NeurWIN  N = 100 M = 25  |  -8080.995 |  -4640.411  |  -2595.329   | -2392.864 |   -2274.130 |   -2261.513 |  -2237.994 | -2253.590 |
> > | WIBQL  N = 100 M = 25  | -6099.899 | -3210.488   |  -3323.359  |  -3310.685 | -3209.533  | -3187.000  |  -3119.380 |-3124.134 |
> >
> >
> > **Recovering bandits**
> >
> > |Policy | episode 0  | episode 5000  | episode 10000  | episode 15000  | episode 20000  | episode 25000  | episode 30000  |
> > | ------------- |-------------| -------------| -------------|-------------|-------------|-------------|-------------|
> > | NeurWIN  N = 4 M = 1 |  172.380  | 573.976| 572.554|575.484 | 575.484| 575.484 | 575.484|
> > | WIBQL  N = 4 M = 1   | 172.380   |556.241 | 564.276| 570.184 |547.794 | 575.484 | 573.487|
> > | NeurWIN   N = 10 M = 1  | 172.380    | 715.302 |716.900 |726.574 |736.475 |736.037 | 736.880|
> > | WIBQL    N = 10 M = 1   |  172.380   |723.084 |735.288 |731.328 |734.410|736.034 |725.965 |
> > | NeurWIN  N = 100 M = 25  | 6335.725    |14349.398 |14313.836 |14387.088 |14387.088 |14387.088 |14387.088 |
> > | WIBQL   N = 100 M = 25  |  6335.725   |13542.083 |13648.613 |13584.329 |13675.496 |13768.953 |13799.544 |
> >
> > ---
> >
> > ### Medium comment 4: "how well does 20-step lookahead perform on (4,1) and (10,1)?"
> >  We have done 20-lookahead for (4,1) and (10,1) for the original setting: for (4,1), 20-lookahead = 575.235, and for (10,1), 20-lookahead = 731.844.
> >
> > 20-lookahead performs better than 8-lookahead but still worse than NeurWIN (N = 4: 575.484, N = 10: 737.811). The reason is that 8-lookahead finds the exact optimal choice within only 8 steps, while 20-lookahead finds the best choice within 2^20 greedily selected choices. As for the performance of QWIC, Since both QWIC and 8-lookahead are heuristics, we are not sure much can be said about the relative performance between them.
> >
> > ---
> >
> > ### Medium comment 5: "authors of [17] implement UCB and Thompson-Sampling... it would be helpful to see how NeurWIN compares ... on at least the recovering bandits environment"
> >
> > We used the publicly available reference code for UCB and TS from [17] and integrated it into our setting for the recovering bandits' case. We note that we ran N = 100 M = 25 for only d = 1 since a larger lookahead value gives an intractable runtime. More formally, the policy would have a complexity of $O((N)^{M \times d})$.
> >
> >
> > |lookahead d | UCB  N = 4 M=1 | UCB  N = 10 M=1  | UCB  N = 100 M=25 |
> > | ------------- |-------------| -------------| -------------|
> > | d=1           |   440.424   |  428.011     |   10972.698  |
> > | d=4           |   547.667   |  556.618     |     N\A      |
> >
> > |lookahead d | TS  N = 4 M=1 | TS  N = 10 M=1  | TS  N = 100 M=25 |
> > | ------------- |-------------| -------------| -------------|
> > | d=1           | 306.856     | 309.300      |   7398.398   |
> > | d=4           | 537.043     | 551.816      |     N\A      |
> >
> > NeurWIN outperforms both the lookahead oracles and UCB policies. UCB and TS underperform since the prior distributions of activated arms are updated incrementally, before optimal greedy decisions can be made. Lookahead oracles perform better than UCB and TS since they can greedily activate arms up to d-timesteps starting from t=0.
> >
> > ---
> >
> > ### Medium comments 6/7: "...more context would need to be given about what is meant by "most""
> > ### "some things that could help ... a) Adapt/reframe the numerical "strong indexability" verification step"
> >
> > We thank the reviewer for the comment. We will revise the sentence to be "considered restless bandit problems in the literature" and provide citations to the appropriate works for better accuracy. An automatic checker for indexability is indeed an interesting idea, and is effectively what we used to verify the condition empirically.  It is probably out of scope for this paper, but we will provide the code of our empirical verification in case someone would like to try it on their own problems.  We will also discuss this as a promising future direction as a verification step in providing theoretical analysis of non-strongly indexable cases.
> >
> > ---
> >
> > ### Minor comments:
> > We thank the reviewer for the detailed remarks. We will incorporate all paper structuring, suggestions, and sentence rephrasing in the revised paper.

---

### Official Review · Reviewer_Ha51 · 2021-07-17

**Rating:** 7
**Confidence:** 3

**Summary:**

The paper proposes a method to automatically learn the Whittle Indices for a Restless Multi-Armed Bandit (RMAB) problem.

Contributions:

The Algorithm: The algorithm tries to learn a neural network that takes as input the state of a given arm and provides as output the whittle index of that state. It does this by showing a connection between the whittle indices for an arm and the indices of an optimal index policy for a family of MDPs (Env(𝝀)) based on that arm’s MDP (Env). It then learns a good index policy for this family of MDPs using a REINFORCE-type method.

Theoretical Justification: Towards substantiating the claim that learning a good policy for Env(𝝀) is equivalent to learning a good whittle index, the paper provides a proof about an epsilon-delta relationship between learning a good policy and a good whittle index.

Empirical Justification: The paper shows that the proposed algorithm works well on 3 previously published RMAB instances, when compared with 1 similar ‘learning a whittle index’-type baseline and 2 reinforcement learning baselines.


**Limitations And Societal Impact:**

Limitations: While the paper does discuss possible limitations of its work, the analysis is not of high quality.

Societal Impact: The paper details an algorithm for a general class of problems. From its description, there doesn’t seem to be any obvious ethical impacts that would be associated with its usage across the broad class of problems. However, those considering employing this algorithm for a specific application should perform an application-specific analysis.


**Main Review:**

Positives:
1) Interesting idea: There has been a lot of recent interest in trying to automatically calculate the whittle index. However, to my knowledge, the approach outlined in this paper is different from those presented in past/contemporary work. Here, the algorithm tries to learn the whittle indices for an arm by learning a good index policy for a superset of MDPs (Env(𝝀)) in which you have to pay a cost of 𝜆 to choose the ‘active’ action. This is because the indices of such an optimal index policy are equal to the whittle indices (under indexability assumptions), and even if you can’t learn the optimal policy, the paper has a proof that suggests an almost optimal policy leads to almost correct whittle indices. There are lots of good algorithms for solving MDPs and so reducing the whittle index problem to this new one seems like a good idea.

2) Experiments on previously published RMAB instances: Rather than arbitrarily creating RMAB instances, the paper shows the effectiveness of the proposed algorithm on 3 classes of RMABs that have been previously published in the literature and, as a result, are more likely to be relevant to practice. In addition, the experimental results indicate that this method reaches a performance level equal to that of the optimal/SOTA solution.

Negatives:
1) Weak baselines and limited related work: While the experimental domains are great,  I believe that the comparison baselines are quite weak. Importantly, the paper fails to include WIBQL [1], a provably correct Q-Learning based algorithm for automatically learning whittle indices. Although the exact implementation presented in that paper is for the average reward setting and tabular case, there is a fairly straightforward extension that would allow you to learn a neural network-based function approximator for these indices. Even without this, I believe that the RL-based baselines are too weak and instead simpler baselines like decomposing the joint problem into individual problems would work better. (To do this, you’d solve the MDP to get the optimal Q-values for each arm independently, and then choose the M arms with the greatest Q(s[t], active)-Q(s[t], passive).)

2) Weak analysis of limitations:
While the paper acknowledges the fact that strong indexability is important to the performance of this method, it doesn’t test the performance of the method on RMAB instances that aren’t strongly indexable. Instead, it makes the strange claim that, because the 3 chosen instances are strongly indexable, ‘most practical bandits problems are indeed indexable’. The conclusion that I draw from this fact is that ‘people publish papers about RMABs that have nice properties’. It would be useful to see the results of the algorithm on RMAB instances that aren’t indexable.
The paper recognizes that it is important to have a simulator for this algorithm to work (because of its need for on-policy rollouts). However, in defending it, the paper argues that the algorithm still performs well on a slightly worse simulator. While this is a useful experiment in showing the algorithm’s robustness to noise, it does not address the underlying limitation. It would be more interesting if the authors commented on possible offline/off-policy extensions to their proposed algorithm.

3) Proof is nice to have, but there are issues: While it’s useful to have a proof that shows some relationship between the closeness of the indices of a good index policy for Env(𝝀) and the true whittle indices, (a) I don’t understand the statement of the proof, and (b) it’s quite weak. For (a), there are 2 sets of if-then relationships in the statement which make it hard to understand. In addition, it’s not clear how  is related to  and -- for example, for something to be 0.1-accurate, what are the conditions on  and . Also, the statement seems to be a bit different from the normal - definition; wouldn’t the statement always be trivially true for an arbitrarily large ? It would be useful if you could clarify the statement and provide a proof sketch. For (b), the proof doesn’t say anything quantitative. To begin with, it’s not clear how  and  are related, i.e., how close to the optimal whittle indices you’ll get for being within an \epsilon ball of the optimal. Even if it’s small, it’s unlikely that you will ever know what the optimal is and, as a result, how close you are to it. Finally, even if you did know the distance between your estimate of the whittle index and the true index, it’s unclear what impact that would have on the final expected return for the RMAB (across multiple arms).

Other Comments:
1) Paper organisation could be improved.
A proof sketch for Theorem 3 would be nice to have in the main text given that it’s the primary theoretical contribution.
Section 3.3 does not add much value and could be pushed to the appendix in favor of the proof sketch.
The details of the NeurWIN algorithm could be introduced earlier; they’re currently at the end of page 5 (more than half way through the paper).
2) The neural network used is very small (16, 32 dimensional layers). Do things change if you use bigger/smaller neural networks/different architectures?
3) In Figure 6, it seems like NeurWIN does better w.r.t. baselines for the Recovering Bandits and Wireless Scheduling problems when there is a noisy simulator vs. the exact simulator. It’s not clear why that should be the case.
4) There is some contemporaneous work on this problem (cited below) that the authors might find interesting.

Conclusion: The idea is interesting and the experimental results seem to show that the algorithm does its job in practice. As a result, despite a perfunctory limitations section, weak baselines and unclear theoretical contributions, I recommend a weak accept for the paper. I would increase my rating if the authors address the comments about the limitations and include stronger baselines.

Citations:
[1] Avrachenkov, Konstantin, and Vivek S. Borkar. "Whittle index based Q-learning for restless bandits with average reward." arXiv preprint arXiv:2004.14427 (2020).
[2] Biswas, Arpita, et al. "Learn to Intervene: An Adaptive Learning Policy for Restless Bandits in Application to Preventive Healthcare." arXiv preprint arXiv:2105.07965 (2021).

---
EDIT AFTER REBUTTAL: Increased score from 6 to 7 (see response to rebuttal for more details).


**Time Spent Reviewing:**

7

---

> ### Author Response · Authors · 2021-08-10
> **response (1/2)**
>
>
> We thank the reviewer for their feedback. We provide responses for outlined negatives:
>
> ### 1. "Importantly, the paper fails to include WIBQL, a provably correct Q-learning based algorithm"
>
> We have implemented the WIBQL algorithm in tabular case, and apply it to the deadline for 50 runs, and the recovering bandits' settings.
> We provide numerical results and explanation below:
>
> **Deadline scheduling**
>
> |Policy | episode 0 | episode 200 | episode 400   | episode 500  | episode 800  | episode 1000 | episode 1500  | episode 2000  |
> | ------------- |-------------| -------------| -------------|-------------|-------------| -------------|-------------|-------------|
> | NeurWIN  N = 4 M = 1 |  -301.034   |  -181.607  |  -116.450 |  -108.173 |  -100.366  | -99.768  | -99.472 |-99.250 |
> | WIBQL   N = 4 M = 1  |   -241.982  | -153.848  | -130.753  | -121.844 |  -118.412 | -126.747   | -115.395 |-112.080 |
> | NeurWIN  N = 10 M = 1 |  -829.923  | -675.260  |  -580.247  | -566.072  | -553.248 |  -551.771  | -551.325 |-551.100 |
> | WIBQL   N = 10 M = 1  |  -753.276   | -624.944  |   -613.419  | -609.431 |  -616.215  | -607.776 |  -598.347 |-592.125 |
> | NeurWIN  N = 100 M = 25  |  -8080.995 |  -4640.411  |  -2595.329   | -2392.864 |   -2274.130 |   -2261.513 |  -2237.994 | -2253.590 |
> | WIBQL  N = 100 M = 25  | -6099.899 | -3210.488   |  -3323.359  |  -3310.685 | -3209.533  | -3187.000  |  -3119.380 |-3124.134 |
>
>
> **Recovering bandits**
>
> |Policy | episode 0  | episode 5000  | episode 10000  | episode 15000  | episode 20000  | episode 25000  | episode 30000  |
> | ------------- |-------------| -------------| -------------|-------------|-------------|-------------|-------------|
> | NeurWIN  N = 4 M = 1 |  172.380  | 573.976| 572.554|575.484 | 575.484| 575.484 | 575.484|
> | WIBQL  N = 4 M = 1   | 172.380   |556.241 | 564.276| 570.184 |547.794 | 575.484 | 573.487|
> | NeurWIN   N = 10 M = 1  | 172.380    | 715.302 |716.900 |726.574 |736.475 |736.037 | 736.880|
> | WIBQL    N = 10 M = 1   |  172.380   |723.084 |735.288 |731.328 |734.410|736.034 |725.965 |
> | NeurWIN  N = 100 M = 25  | 6335.725    |14349.398 |14313.836 |14387.088 |14387.088 |14387.088 |14387.088 |
> | WIBQL   N = 100 M = 25  |  6335.725   |13542.083 |13648.613 |13584.329 |13675.496 |13768.953 |13799.544 |
>
>
> From the results, NeurWIN achieves a better performance compared to WIBQL especially with larger states spaces (120 per arm for deadline scheduling and 20 per arm for recovering bandits), while WIBQL performance degrades with larger state spaces. With more arms, WIBQL updates an arm's Q table less frequently and requires more timesteps to sample all arms to reach $\epsilon$- optimal Q values, which explains the degraded performance as more arms are in the system.
> We also tried implementing WIBQL for the wireless scheduling case, but a tabular RL method failed to run for 2 million states per arm.
>
> In summary, WIBQL performed well in small state spaces (recovering bandits), poorly compared to NeurWIN in a larger state space (deadline scheduling), and failed to run for wireless scheduling case with 2 million states per arm. The testing parameters are the same as described in section 5.1 ($\beta$ = 0.99 , averaged over 50 runs for deadline scheduling, episode horizon = 300 timesteps).
>
> ---
>
> ### 2. "It would be more interesting if the authors commented on possible offline/off-policy extensions to their proposed algorithm."
>
> We agree with the reviewer that extending NeurWIN to the offline or off-policy settings would be a promising research direction.
> For NeurWIN, one way to utilize the data samples collected offline or via a behavior policy is to first construct a predictive model for each restless arm based on these historical data and then learn the Whittle index by the NeurWIN training algorithm based on the predictive model.
> In the context of general MDPs, similar attempts of offline RL have recently been made by [Kidambi et al., 2020] and [Yu et al., 2020].
> The estimation error of the predictive model can be quantified in a way similar to the prior works on RL with a generative model (e.g., [Azar et al., 2012]). Compared to RL for general MDPs, it is expected that NeurWIN would require less offline data to learn the predictive model due to the decomposition provided by the Whittle index.
> We will add one remark to highlight these potential extensions in the final version.
>
> References:
> [Kidambi et al., 2020] Kidambi, Rahul, Aravind Rajeswaran, Praneeth Netrapalli, and Thorsten Joachims. “MOReL: Model-Based Offline Reinforcement Learning,” NeurIPS 2020
>
> [Yu et al., 2020] Yu, Tianhe, Garrett Thomas, Lantao Yu, Stefano Ermon, James Zou, Sergey Levine, Chelsea Finn, and Tengyu Ma. "MOPO: Model-Based Offline Policy Optimization." NeurIPS 2020.
>
> [Azar et al., 2012] Azar, Mohammad Gheshlaghi, Rémi Munos, and Bert Kappen. "On the Sample Complexity of Reinforcement Learning with a Generative Model." ICML 2012.
>
>
> For non-strongly indexable arms, a preprocessor step for checking the strong indexability can be added as suggested by reviewer jp8H.
> Our goal in this paper was to compare learned policy against known indexable cases from the literature, and it would be a promising future direction to provide theoretical analysis of NeurWIN on non-strongly indexable arms.

---

> > ### Author Response · Authors · 2021-08-10
> > **response (2/2)**
> >
> > ### 3. "Proof is nice to have, but there are issues"
> >
> > The issue of clarity was raised by other reviewers as well, and we provide here a revised version of theorem 3 and associated definitions, which we hope will clarify the results:
> >
> > Consider a neural network with parameter $\theta$. Let $ \tilde{Q_\theta}(s, \lambda)$ be the average reward of applying this neural network to Env($\lambda$) with initial state $s$. If the neural network is indeed the optimal controller to Env($\lambda$), then we have $\tilde{Q_\theta}(s, \lambda)= \max\\{Q_{\lambda, act}(s), Q_{\lambda, pass}(s)\\}$, for all $s$ and $\lambda$. We further define the near-optimality of the neural network as follows:
> >
> > Definition: A neural network with parameter $\theta$ is said to be $\epsilon$-optimal if there exists a positive number $\delta$ such that $\tilde{Q_\theta}(s_1, \lambda)\geq \max\\{Q_{\lambda, act}(s_1), Q_{\lambda, pass}(s_1)\\}-\epsilon$, for all $s_0, s_1$ and $\lambda\in [f_\theta(s_0)-\delta, f_{\theta}(s_0)+\delta]$.
> >
> > We then show that a near-optimal neural network is also Whittle-accurate.
> >
> > Revised Theorem 3: If the arm is strongly indexable, then for any $\gamma>0$, there exists a positive $\epsilon$ such that every $\epsilon$-optimal neural network is also $\gamma$-Whittle-accurate.
> >
> > Based on the revised Theorem 3, our goal is to train a near-optimal neural network. We define our objective function as $\sum_{s_0, s_1} \tilde{Q}_\theta(s_1, f_\theta(s_0))$. In each iteration, we randomly choose $s_0, s_1$ and aim to update $\theta$ based on $\nabla_\theta \tilde{Q}_\theta(s_1, f_\theta(s_0))$. To this end, we employ the REINFORCE algorithm as described in Alg. 1.
> >
> >
> > ---
> >
> > ## Other comments' response:
> >
> > ### 1. "paper organization could be improved"
> >
> > Section 3.3 will be pushed to the appendices as recommended, and will be replaced with the revised theorem.  We will also update the paper to include an earlier description of NeurWIN in the main text.
> >
> > ---
> >
> > ### 2. "neural network used is very small... do things change if you use bigger/smaller neural networks..."
> >
> > We have retrained NeurWIN for a smaller {8, 14} network with 165 parameters per arm, and larger {48, 64} network with 3345 parameters per arm. Original network has 625 parameters per arm for deadline and wireless scheduling cases, and 609 parameters for recovering bandits. Networks have two hidden fully connected layers.
> > We use the same training parameters as in the original network ($\beta$ = 0.99, episode horizon = 300 timesteps).
> > We observe that the smaller network requires in general more episodes to reach the total discounted rewards as the original network.
> > The larger network requires less episodes compared with the original network to reach the same total discounted rewards.
> > We also note that in recovering bandits N = 10  M = 1, the smaller network failed to reach the same total discounted rewards as the original network, and performed worse than the 8-lookahead policy.
> >
> > For example, in deadline scheduling N = 4 M = 1 the smaller network requires 1380 episode to reach the Whittle index performance. The larger network in contrast reaches the Whittle index performance with considerably fewer episodes at 180 episodes, while the original network requires 600 episodes.
> > For N = 10, M = 1: smaller network requires 1400 episodes to reach Whittle index performance. Bigger network requires 200 episodes.
> > For N = 100, M = 25: smaller network requires 1410 episodes, with the bigger network needing 305 episodes to reach the Whittle index performance in terms of total discounted rewards.
> >
> > We provide some numerical results for the three cases:
> >
> > **Deadline scheduling**
> >
> > |Policy | episode 0  | episode 200 | episode 400 | episode 500  | episode 1000  | episode 1500  | episode 2000  |
> > | ------------- |-------------| -------------| -------------|-------------|-------------|-------------|-------------|
> > | N = 4 M = 1  smaller | -270.603 |  -191.148 | -150.413 | -144.548 | -117.171 | -104.102 | -102.922 |
> > | N = 4 M = 1  larger | -265.290 | -103.801 | -101.731 | -100.349 |-100.642| -99.725 |-99.852 |
> > | N = 10 M = 1   smaller   | -813.355 | -667.187 | -623.406 | -617.207| -579.997 | -560.114 | -558.709 |
> > | N = 10 M = 1   larger  | -808.657 | -560.358 | -556.656 | -553.116 | -552.399 | -552.191 | -552.252 |
> > | N = 100 M = 25  smaller   | -7271.401 | -4735.629 |  -3657.709 | -3477.199 |-2547.508 | -2309.878| -2296.605|
> > | N = 100 M = 25  larger    | -7165.802 | -2306.808 | -2269.159 | -2261.722 | -2230.311 | -2234.388 | -2230.423 |
> >
> >
> > **Recovering bandits**
> >
> > |Policy | episode 0  | episode 5000  | episode 10000  | episode 15000  | episode 20000  | episode 25000  | episode 30000  |
> > | ------------- |-------------| -------------| -------------|-------------|-------------|-------------|-------------|
> > | N = 4 M = 1  smaller |  172.380   | 573.231 | 573.594 |573.976 |575.091 | 575.014| 574.085 |
> > | N = 4 M = 1  larger |   172.380  |575.484 | 575.484 |575.484 |575.484 |575.484 | 575.484|
> > | N = 10 M = 1   smaller  | 172.380  | 714.658 | 715.439 | 716.368 | 716.614 | 719.896 | 721.152 |
> > | N = 10 M = 1   larger |   172.380  | 726.649| 736.846| 736.233|736.888 |730.062 | 736.531|
> > | N = 100 M = 25  smaller  |  6335.725   | 14330.777 | 14339.849 | 14349.398 | 14377.269 |14375.343 | 14352.127 |
> > | N = 100 M = 25  larger   |   6335.725  | 14387.088|14387.088 | 14387.088| 14387.088|14387.088 | 14387.088|
> >
> >
> > **Wireless scheduling**
> >
> > |Policy | episode 0  | episode 5000  | episode 10000  | episode 15000  | episode 20000  | episode 25000  | episode 30000  |
> > | ------------- |-------------| -------------| -------------|-------------|-------------|-------------|-------------|
> > | N = 4 M = 1  smaller |  -261.195   |-180.690 | -183.748 | -184.738 |-185.271 |-185.166 | -185.768 |
> > | N = 4 M = 1  larger |   -323.224   |-183.893 |  -185.132| -184.932| -183.102| -184.468| -184.511|
> > | N = 10 M = 1   smaller  |  -748.777   | -598.003 |-564.955 |-571.566 |-566.231 | -572.509 |-561.637 |
> > | N = 10 M = 1   larger | -950.959    |-565.319 |-561.604 |-565.187 | -566.350| -556.023 | -564.717|
> > | N = 100 M = 25  smaller  |  -5815.304   | -3472.804 |-3446.648 |-3500.983 |-3497.883 |-3566.333 | -3469.641|
> > | N = 100 M = 25  larger  |  -8284.063   |-3521.415 |-3441.050 |-3504.629 | -3548.891 | -3466.376|-3530.511 |
> >
> > ---
> >
> > ### 3. "In figure 6, it seems like NeurWIN does better w.r.t. baselines ... when there is a noisy simulator"
> >
> > With added noise, we observed that NeurWIN learns an index different from the true Whittle index, but still preserves the state ordering given by the true Whittle index.
> > In other words, for two states $s_1, s_2$, the learned indices are $f_\theta(s_1) > f_\theta(s_2)$ if $W(s_1) > W(s_2)$, and an arm in state $s_1$ would still be selected over an arm in state $s_2$.
> > In this sense, the strong indexability condition of strictly decreasing discounted net reward still holds but for a different set of activation costs.
> > In recovering bandits specifically, the added noise generated a different state ordering than that of the true Whittle index, which leads to degraded performance in terms of total discounted rewards.
> > We will update the paper to include this description.

---

> > > ### Comment · Reviewer_Ha51 · 2021-08-23
> > > **Response to Rebuttal**
> > >
> > > Comments:
> > >
> > > **WIBQL Baseline:**
> > > Thank you for implementing the WIBQL baseline. While I commend the authors for their effort, I disagree with their conclusions. Specifically, the takeaway seems to be that it’s not clear whether NeurWIN’s ability to perform better than WIBQL on medium to large instances is because of the underlying algorithm or NeurWIN’s neural network function approximation for the Q values. While the algorithm presented in Avrachenkov, et. al. is for the tabular case, it’s not complicated to extend it to the case where you have a neural network function approximator for the Q and $\lambda$ values. As a result, saying NeurWIN is more scalable than WIBQL doesn't seem convincing to me.
> > >
> > > *(Aside: (A) Just like how NNs are not central to WIBQL, I don’t believe NNs or even REINFORCE is central to NeurWIN’s approach to finding whittle indices. It would be interesting to see how well NeurWIN performs in the tabular case or with an actor-crtic algorithm in which the actor network is the current implementation of NeurWIN. (B) Having worked with WIBQL previously, I know that tuning the learning rate is very important. From the data presented, it seems like the learning rate for learning lambda may be low, leading to slow convergence. It would be useful to include information about hyperparameters used with the WIBQL baseline.)*
> > >
> > > **Limitations:**
> > >  While the authors’ comment that the application of NeurWIN to non-indexable problems is outside the scope of the paper is well taken, the fact that it is (currently) only shown to be applicable to strongly indexable problems is a limitation. Rather than claim that `all interesting/important problems are strongly indexable’ as is done in the paper, I believe that (a) making it clear that this is a limitation and then (b) suggesting strategies (like that suggested by reviewer jp8H) is more appropriate. More broadly, I believe that the limitations section from the submission skirted around the actual limitations of the algorithm and the paper would be better served by explicitly acknowledging them.
> > >
> > > **Proof:**
> > > The re-written theorem is more readable and I think I understand what the authors are trying to say, however, I’m still not sure I fully understand it as it is written. You say ‘there exists a positive $\epsilon$ such that every $\epsilon$-optimal neural network is also $\gamma$-Whittle-accurate’. What’s stopping this from being trivially true for $\epsilon=0$ (similarly with $\delta=0$ in the definition of $\epsilon$-accurateness)?
> > >
> > > **Size vs. Performance:** It’s very interesting to see how increasing the size improves the quality of results. Given these benefits, it would be great if the authors include a more detailed analysis of the ‘size vs performance’ tradeoff to see how these gains scale.
> > >
> > > **Conclusion:** Despite the criticisms made above, I believe the authors have addressed my key concerns to a fair extent. While I’m still not sure of the theoretical contribution, I think that implementing WIBQL (at least as described in the original paper) and addressing my concern about the limitations of the simulator satisfactorily meet my requirements for increasing the rating of the paper. To reflect this I will increase my score from 6 to 7.

---

### Decision · Program_Chairs · 2021-09-27

**Decision:**

Accept (Poster)

**Comment:**

The paper studies the important problem of restless bandit. While the Whittle index is known to optimally solve the problem, it remains hard to compute in many practical problems. The authors propose a solution leveraging the generalization capabilities of neural networks in a non trivial way and obtains a practically feasible and accurate method.

After the rebuttal and the discussion, the reviewers agree that the problem is significant and the contribution is interesting and it is likely to set an important baseline for future work on the topic. For this reason I recommend acceptance. Yet, there remains a number of aspects that the authors should improve in the final version:
- Add the comparison to WIBQL detailing how it is implemented and clarifying that defining a scalable version (i.e., with neural networks) is outside the scope of the paper.
- Add the improved version of the theory and possibly include an additional paragraph discussing in detail its relevance.
- Clearly acknowledge the need for a simulator and discuss how this could be possibly relaxed as a future work.
- Integrate reviewers' suggestions while improving the structure and overall clarity of the paper.